# Prognosis in Chronic Myeloid Leukemia: Baseline Factors, Dynamic Risk Assessment and Novel Insights

**DOI:** 10.3390/cells12131703

**Published:** 2023-06-23

**Authors:** Miriam Iezza, Sofia Cortesi, Emanuela Ottaviani, Manuela Mancini, Claudia Venturi, Cecilia Monaldi, Sara De Santis, Nicoletta Testoni, Simona Soverini, Gianantonio Rosti, Michele Cavo, Fausto Castagnetti

**Affiliations:** 1Dipartimento di Scienze Mediche e Chirurgiche (DIMEC), Università di Bologna, 40138 Bologna, Italy; sofia.cortesi2@studio.unibo.it (S.C.); cecilia.monaldi2@unibo.it (C.M.); sara.desantis9@unibo.it (S.D.S.); nicoletta.testoni@unibo.it (N.T.); simona.soverini@unibo.it (S.S.); michele.cavo@unibo.it (M.C.); fausto.castagnetti@unibo.it (F.C.); 2Istituto di Ematologia “Seràgnoli”, IRCCS Azienda Ospedaliero-Universitaria di Bologna, 40138 Bologna, Italy; emanuela.ottaviani@aosp.bo.it (E.O.); mancini_manu@yahoo.com (M.M.); claudia.venturi2@studio.unibo.it (C.V.); 3Istituto Scientifico Romagnolo per lo Studio e la Cura dei Tumori (IRST) IRCCS “Dino Amadori”, 47014 Meldola, Italy; gianantonio.rosti@unibo.it

**Keywords:** chronic myeloid leukemia, prognosis, risk assessment, genomic factors

## Abstract

The introduction of tyrosine kinase inhibitors (TKIs) has changed the treatment paradigm of chronic myeloid leukemia (CML), leading to a dramatic improvement of the outcome of CML patients, who now have a nearly normal life expectancy and, in some selected cases, the possibility of aiming for the more ambitious goal of treatment-free remission (TFR). However, the minority of patients who fail treatment and progress from chronic phase (CP) to accelerated phase (AP) and blast phase (BP) still have a relatively poor prognosis. The identification of predictive elements enabling a prompt recognition of patients at higher risk of progression still remains among the priorities in the field of CML management. Currently, the baseline risk is assessed using simple clinical and hematologic parameters, other than evaluating the presence of additional chromosomal abnormalities (ACAs), especially those at “high-risk”. Beyond the onset, a re-evaluation of the risk status is mandatory, monitoring the response to TKI treatment. Moreover, novel critical insights are emerging into the role of genomic factors, present at diagnosis or evolving on therapy. This review presents the current knowledge regarding prognostic factors in CML and their potential role for an improved risk classification and a subsequent enhancement of therapeutic decisions and disease management.

## 1. Introduction

Chronic myeloid leukemia (CML) is a clonal myeloproliferative disorder characterized by a reciprocal t(9;22)(q34;q11) translocation, designated the Philadelphia (Ph) chromosome [1]. The resulting *BCR::ABL1* fusion gene on the Ph chromosome encodes the chimeric BCR::ABL1 protein, a constitutively active tyrosine kinase [2,3] driving the initiation and maintenance of the disease [4,5]. The vast majority of CML patients are diagnosed in chronic phase (CP), but, if untreated, all patients inevitably progress to blast phase (BP, possibly through an intermediate accelerated phase, AP; Table 1) [6,7,8,9] and eventually death. As a result of the discovery of these cytogenetic and molecular hallmarks of CML, there was a major therapeutic breakthrough with the development of a targeted therapy, the tyrosine kinase inhibitors (TKIs). Firstly, imatinib was introduced in early 2000 [10], followed by newer more potent TKIs of second-generation (2G) (nilotinib, dasatinib and bosutinib) and third-generation (3G) (ponatinib), associated with the achievement of faster and deeper responses and a lower risk of disease transformation, although in the absence of a significant survival benefit compared with imatinib [11]. Thus, the rates of progression have been reduced to 0.5–2% per year from more than 10–20% per year in the pre-TKI era [12].

The significant expansion of our knowledge of the biological mechanisms that underlie the disease evolution has helped to refine treatment decisions and response monitoring, contributing to the dramatic improvement of the outcome of CML patients [13,14,15,16], who currently have a life expectancy approaching that of the general population [17]. Furthermore, some selected cases may successfully discontinue TKI treatment, making treatment-free remission (TFR) a new goal to pursue according to the most recent guidelines [18,19,20].

However, there is still a minority of cases who fail TKI treatment and progress from CP to advanced disease, requiring a more aggressive therapeutic approach where allogeneic stem-cell transplantation (allo-SCT) still has a fundamental role [21,22]; prognosis of transplant-ineligible patients remains particularly dismal, with a median survival usually less than one year [21]. Although there is continuous evolution of the therapeutic landscape and management goals in CML, the prevention of the leukemic transformation and a normal survival clearly remain the main purposes [18]. To this end, the early recognition of the risk of treatment failure and progression in patients with non-advanced phase CML is extremely relevant [18,19,20].

In this review we will summarize the prognostic factors in CML, some well-known and well-established and others more recently identified and not fully validated, potentially useful for a customized therapeutic approach, aimed at selecting the most appropriate TKI taking into account the profile and aggressiveness of each CP-CML patient’s disease.

## 2. Baseline Prognostic Factors

### 2.1. Age and Comorbidities

CML is a disease of the elderly, with a median age at diagnosis of about 65 years [23]. The negative prognostic role of older age was well recognized in the pre-TKI era. In particular, the elderly experienced more toxicities with interferon-alpha (IFN-α) therapy, resulting in poorer treatment tolerance and inadequate drug delivery and significantly higher rates of transplantation-related mortality regardless of the underlying disease [24,25,26]. Many studies revealed that the negative impact of older age on response, and, partially, survival rates were nearly eliminated with the introduction of TKIs [27,28,29,30,31]. The MD Anderson Cancer Center (MDACC) group [27] conducted the first extended analysis on efficacy and safety of imatinib in older patients, showing similar cytogenetic response rates compared to younger CP patients. To follow, European experiences, in clinical trials and real-life settings [28,29,30,32], confirmed the absence of significant differences between elderly and younger patients treated with frontline imatinib as to cytogenetic/molecular responses and long-term outcomes. More recently, a similar efficacy was described for 2G-TKIs both in elderly and younger patients [30,31].

On the contrary, adolescents and young adults (AYAs) represent a small group of CML patients in which the disease shows features of greater aggressiveness similarly to pediatric population, in particular a larger spleen size at diagnosis, with a worse prognostic profile in terms of short-time outcome. In spite of poorer prognostic indicators, lower response rates and higher probability of early disease progression of the AYAs, no inferior progression-free survival (PFS) and overall survival (OS) were actually observed between the various age groups [32,33,34,35].

Although age per se does not represent an adverse prognostic factor, multimorbidity becomes more common with aging [36]. Treatment optimization with any TKI, both in terms of selection and dose adjustment of the drug, is consequently challenged by the higher rate of adverse events reported in elderly patients. This is particularly relevant with newer-generation TKIs, associated with a higher risk of cardiovascular events [37,38] that now are the main cause of death in CP-CML patients just as in the general population [39]. The Charlson Comorbidity Index (CCI) is a common and validated tool for evaluating the impact of relevant comorbid conditions on life expectancies [40]. In the German CML Study IV [41], a randomized trial designed to optimize imatinib therapy, higher CCI at diagnosis was significantly associated with lower OS probabilities: the 8-year OS probabilities for patients with CCI 2, 3 to 4, 5 to 6 and ≥7 were 93.6%, 89.4%, 77.6% and 46.4%, respectively. No differences in cumulative incidences of AP, BP or remission rates were observed between patients in different CCI groups. In a multivariate analysis, CCI was the most powerful predictor of survival. Similar results were observed also in patients treated with 2G-TKIs [42,43,44] or using different comorbidity scoring systems [45]. 

### 2.2. Prognostic Models

Since the introduction of the first therapeutic weapons, there has been an increasing effort to detect factors that could estimate CML patients’ prognosis at diagnosis in order to select the most appropriate treatment option. Many risk scores have been developed using simple clinical and hematologic baseline parameters (Table 2) [18]. 

In the early eighties, when CML was treated only with conventional chemotherapy (mainly busulfan and hydroxyurea), the first prognostic system introduced was the Sokal score [46] that, based on four baseline variables (patient’s age, spleen size, platelet count and percentage of blast cells in the peripheral blood), stratifies patients into three risk categories (low, intermediate and high) with significantly different OS probabilities. In the TKI era, the Sokal score still remains the most popular and a worldwide-used score.

Later, in 1998, a new scoring system was developed for estimation of survival in CML patients treated with IFN-α, the Hasford or Euro score [47], that considers the same parameters plus basophil and eosinophil counts at diagnosis, providing a better discrimination between the three risk categories compared to the Sokal score in this subset of patients.

A further step forward in this field was felt rational and necessary with the revolutionary advent of TKIs. In 2011, the ELN (European LeukemiaNet) published the European Treatment and Outcome study (EUTOS) score [48], developed and validated using data of patients treated with first-line imatinib to predict the complete cytogenetic response (CCyR) status at 18 months, regarded as the most robust surrogate marker of long-term survival. This score, based only on two parameters (namely, the percentage of basophils in the blood and spleen size), showed to best discriminate between low- and high-risk groups of patients.

Given the improved survival induced by imatinib and the availability of large sample and long-term data, in 2016, a new EUTOS Long Term Survival (ELTS) score was designed in order to predict the risk of dying because of CML (“CML-related death” or “leukemia-related death”, LRD), defined as death after disease progression [49]. Only the new ELTS score, compared to the other available prognostic models, proved to recognize three significantly different risk groups regarding cumulative incidence probabilities of CML-specific death and survival probabilities. Further studies confirmed that the ELTS score outperformed the other scores, demonstrating its superior predicting value regarding not only LRD but also molecular responses, PFS and OS, especially in patients receiving 2G-TKIs [50,51,52,53]. The same four parameters included in the Sokal score were identified in the ELTS score, but with different weights. In particular, the impact of the variable “age” was lower in the ELTS score than in the Sokal score, due to the greater prognostic value of age in the chemotherapy era. This translated into a different patient distribution into risk groups: about half of the patients classified as high-risk by the Sokal score were re-allocated as non-high-risk with the ELTS score [50]. An Italian analysis, using data from multicenter studies conducted by the GIMEMA CML Working Party (WP) [54], showed that the number of patients misclassified by the Sokal score was particularly relevant in the elderly group, with a risk of unnecessary over-treatment especially of this category of patients with more frailties. Moreover, in a study from the International Registry for Childhood Chronic Myeloid Leukemia, the ELTS score showed better differentiation of PFS compared to the other scoring systems in children and adolescent with CML, although further studies are required to confirm the applicability of this score also in the pediatric setting [55].

Both the 2020 ELN and 2019 GIMEMA CML WP recommendations strongly suggest the use of the ELTS score as the preferred system to assess prognostic baseline risk [18,19].

### 2.3. Cytogenetic Prognostic Factors

The constitutional activation of BCR::ABL1 and its downstream pathways result in excessive proliferative stimulus and accumulation of DNA damage, probably via oxidative stress [21,56,57]. The subsequent genomic instability of CML cells may explain the possible emergence of additional chromosomal abnormalities (ACAs) other than the distinctive Ph chromosome. The evidence of these clonal chromosomal alterations is synonym of a likely greater aggressiveness of the disease, and their development during treatment (referred to as “clonal evolution”) is confirmed to be a signal of acceleration (see also the paragraph “Prognostic factors beyond the baseline”) [7,58]. In fact, the incidence of these ACAs differs across the disease phases [59,60,61], ranging from 5–10% at diagnosis in CP to 30% in AP and 50–80% in BP [62,63,64,65].

Historically, based on the frequency of the pathways of clonal evolution followed by CML cells during disease evolution, ACAs are subdivided into the more common “major route” (trisomy 8 (+8), i(17q), trisomy 19 (+19) and an extra copy of Ph (+Ph)), each occurring in >10% of cases with ACAs, and “minor route” (such as trisomy 21 (+21), t(3;12), t(4;6), t(2;16), t(1;21), loss of the Y chromosome (−Y)) [66,67]. In the pre-TKI era, the presence of ACAs was more consistently associated with poor prognosis [64,68,69,70]. Data collected since the introduction of imatinib are more conflicting and the exact prognostic implication of ACAs in CML in the context of TKIs is still unclear, in particular at the time of diagnosis in CP.

Two studies evaluating the role of ACAs already present at initial diagnosis in imatinib-treated CP patients suggested a negative prognostic impact. In the GIMEMA CML WP analysis [59], based on data from 21 patients with ACAs at diagnosis out of a total of 378 evaluable CP patients (5.6%), the overall cytogenetic and molecular response rates were significantly lower and the time to response was significantly longer in patients with ACAs; these differences in terms of responses, rate and rapidity, however, did not translate into significant differences in terms of long-term outcomes. The different prognostic significance of the distinct kinds of abnormalities, not assessable in this analysis due to the few available cases, was investigated in the CML Study IV [60]. Of 1151 evaluable patients with CP-CML at diagnosis treated with imatinib-based regimens, 79 patients had ACAs (6.9%); among these, 38 patients (48.1%) had −Y, 25 (31.6%) had minor route ACAs, and 16 (20.3%) had major route ACAs. Only major route ACAs had prognostic relevance with evidence of inferior response rates, longer time to CCyR and major molecular response (MMR) and poorer survival. The 5-year PFS was 90% and 50%, and the 5-year OS was 92% and 53% in the standard t(9;22) and major route ACAs groups, respectively. Neither −Y nor minor route ACAs showed an impact on prognosis in comparison with absence of ACAs, both in terms of times to responses and responses/survival rates. In other studies, however, the adverse effect of ACAs at the time of diagnosis in CP patients treated with TKIs (imatinib or 2G-TKIs) was not confirmed [71,72,73].

ACAs constitute clearly a quite heterogeneous group of cytogenetic abnormalities, and the interpretation of the results may also be confounded by the concurrent presence of multiple chromosome alterations. Moreover, not all ACAs seem to have an equal pathogenetic significance of progression promoters, and some can just be “innocent bystanders” of the genetic instability of CML cells on the way to an advanced disease, where other concurrent factors play a central role [74]. Therefore, the frequency-based stratification of ACAs into major and minor routes does not appear to fully reflect their predictive value. More recently, Wang et al. [61] proposed a revised classification system accounting for the prognostic impact of individual ACAs on treatment response and survival in the TKI era. The authors classified the six more frequently detected single ACAs into two groups based on their impact on TKI treatment response: group 1 included +8, (+Ph), −Y, with a relatively good response; group 2 included i(17q), −7/del(7q), 3q26.2 rearrangements, with a relatively poor response. Consistent with the TKI response pattern, patients with ACAs emerging from CP and at initial CML diagnosis in group 1 had better survival than patients in group 2; when compared to patients with no ACAs, group 1 patients had no significant survival difference. In a following study by Gong et al. [75], the presence of any kind of ACA at diagnosis had actually a significant impact on blastic transformation and a negative predictive role for OS, but particularly the ACAs identified as high-risk ACAs, corresponding to the group 2 of the previous study. The UK SPIRIT2 study [76] found that patients with certain high-risk cytogenetic abnormalities (including the original four major ACAs and five additional lesions: +21, −7/del(7q), 3q26.2, 11q23 abnormalities and complex karyotype) [77] at diagnosis had significantly higher progression rates, worse PFS and freedom-free progression (FFP). Moreover, there was no correlation between Sokal/ELTS scores and the presence/absence of ACAs, and ACAs were significant predictive factors for PFS and FFP, independently of Sokal or ELTS scores.

In relation to the variant Ph chromosome translocations [t(v;22)] (reported in 5–10% of CML patients) and deletions in derivative chromosome 9 [der(9q)], there is a general agreement on regarding them no longer as indicators of poor prognosis in the TKI era [78,79,80].

On the basis of this evidence, current ELN recommendations [18] suggest that patients at diagnosis with high-risk ACAs (+8, (+Ph), i(17q), +19, −7/7q-, 3q26.2 or 11q23 aberrations and complex karyotypes) should be accordingly treated as high-risk patients.

### 2.4. Transcript Type

The location of the breakpoint in the *ABL1* and *BCR* genes is variable, originating distinct isoforms of the resulting BCR::ABL1 chimeric protein. In the great majority of CML patients (~95%), the break in *BCR* occurs within the region between exons 12 and 16 (e12-e16, historically named b1-b5), called the major-breakpoint cluster region (*M-BCR*). Usually, the messenger RNA fusion transcripts are e13a2 (b2a2) and e14a2 (b3a2), which are both translated into the p210^BCR::ABL1^ protein [81,82]. Approximately 5–10% of CML patients co-express both e13a2 and e14a2 transcripts. Because of the possibilities of other breakpoint sites and alternative splicing of the two genes, additional and very rare “atypical” transcript types can be found [81,83]. This evidence has raised questions of “if”, “how” and “why” the different transcript types may influence the outcome of CML patients.

In the pre-TKI era, some studies exploring the potential prognostic impact of the transcript type reported better outcomes conferred by e14a2 transcript, in the absence of robust and definitive conclusions [84,85,86,87]. In the TKI era, although there is more available data, the role played by the transcript type remains to be elucidated.

First of all, no clustering of the different transcript types among the distinct risk categories (as per Sokal, Euro-Hasford, EUTOS and ELTS scores) was observed [88,89,90,91,92,93,94].

In patients who received imatinib (frontline, after or in combination with IFN-α; Table 3), most studies concluded for higher and earlier molecular response rates in favor of e14a2 compared to e13a2 [90,93,95,96,97,98,99,100,101,102,103,104,105,106,107]. Regarding the possible role of the transcript type on the long-term outcomes investigated in some of these studies, there was no evidence of a significant impact in the majority of cases [90,91,92,94,102,105]. In contrast to this, the GIMEMA CML WP [93] observed that the 7-year estimated probabilities of OS, PFS and failure-free survival (FFS) were significantly lower in patients with e13a2 transcript. On the other hand, Pagnano et al. [95] and Marcé et al. [101] found a significant association of the e13a4 with improved OS. Focusing on both OS and probabilities of CML-related death, Pfirrmann et al. [94] showed that the significant survival advantage for patients with the e14a2 disappeared if they were stratified for the groups of the ELTS score.

Few studies were conducted to analyze the prognostic role of transcript type in patients treated with newer-generation TKIs, with more limited and conflicting data (Table 4). Some authors reported better responses for patients expressing e14a2 [91,108,109,110,111,112,113], but others did not find statistically significant differences between the patients expressing e13a2 or e14a2 treated with different TKIs [109,111,113,114,115]. In particular, two studies analyzed separately patients treated with different TKI modalities (imatinib, dasatinib or nilotinib). In the study of Su et al. [111], higher CCyR and MMR rates were found in patients with the e14a2 both in the imatinib- and nilotinib-treated group, but not in the dasatinib-treated group, in the absence of significant differences in the deep molecular response (DMR) rates at 12 months between the two transcripts in all three TKI groups. Jain et al. of the MDACC [91] found that the expression of e14a2 was associated with earlier and deeper responses and higher probabilities of achievement of optimal responses according to the ELN criteria. Among patients bearing e13a2, responses rates (in terms of CCyR, MMR and MR4.5) were lower in those who received imatinib at standard dose compared with other TKI modalities (imatinib at high dose, dasatinib or nilotinib); this trend was not observed in patients with the e14a2, where responses rates were substantially comparable between all TKI groups. Moreover, this is the sole study that found e14a2 as a predictive factor also for improved probability of event-free survival (EFS) and transformation-free survival (TFS). No study demonstrated significant differences in terms of OS between patients with different transcript types treated with different TKIs types.

Two studies evaluated the impact of the transcript type in patients who received only frontline nilotinib with similar finding [112,113]: under nilotinib treatment, patients harboring e14a2 had a significantly higher rate of molecular responses compared to those with the e13a2, in the absence of significant differences in terms of long-term outcomes. The relationship between the transcript type and treatment with the 3G-TKI ponatinib was evaluated in another study from MDACC [116], and, among recurrent/refractory patients, higher rates of CCyR and MMR and a significantly superior 3-year OS were observed in those with the e14a2 transcript.

In almost all citated studies investigating the prognostic impact of the transcript type in the TKI era, patients expressing both e13a2 and e14a2 transcripts had similar (or even better) results to those with the isolated e14a2 transcript. Very little data are available concerning the “atypical” transcripts, having a variable incidence of 0.9–13%, with e1a2 as the most frequent one and typically associated with inferior response to TKIs and worse outcome [117,118,119].

How the difference of only 75 bases and 25 amino acids in the two e13a2/e14a2 transcripts and p210^BCR::ABL1^ proteins, respectively, would reflect into a different clinical impact is still unclear. It may be a quantitative difference of intracellular levels [120,121]; otherwise, there may be a qualitative difference between the two p210^BCR::ABL1^ proteins that could affect the binding site or activity of the TKIs or that could translate into a different ability to elicit an immune response [89,120,122]. Other than biological differences, another possible explanation for the difference in the prognostic impact of the transcript types, especially in terms of molecular responses in the absence of a clear effect on survival, is correlated to technical issues. The real-time quantitative PCR (RT-qPCR) assays employed to measure *BCR::ABL1* mRNA levels appear to be able to amplify the e13a2 amplicon more efficiently than the e14a2 amplicon [120,123]. This discrepancy in amplification performance may be related to the difference in amplicon length generated by the RT-qPCR assay (the e14a2 amplicon is approximately twice as large as e13a2), although the sequence itself may also be important [124,125,126].

**Table 3 cells-12-01703-t003:** Data on response to treatment and long-term outcomes according to transcript type in patients treated only with imatinib-based regimens.

Ref.	TKI	No. of pts	FU	CCyR Rate e13a2 vs. e14a2	MMR Rate e13a2 vs. e14a2	DMR Rate e13a2 vs. e14a2	OS Rate e13a2 vs. e14a2	PFS/TFS/EFS/FFS Rate e13a2 vs. e14a2
Lucas et al. [90]	ima	71	2 y	At 1 y, 25% vs. 54% (*p* = 0.01) At 2 y, 39% vs. 58%	NR	NR	NS	EFS, NS
Hanfstein et al. [105]	ima	1105	5 y	NS	At 5 y, 78% vs. 86% (*p* = 0.002)	MR4.0, 55% vs. 75% (*p* < 0.001)	NS	PFS, NS
Bonifacio et al. [104]	ima	320	6 y	NR	NR	sMR4.0, 53% vs. 63% (*p* = 0.07)	NR	NR
Lin et al. [96]	ima	166	10 y	NR	61% vs. 77% (*p* = 0.02)	NR	NR	NR
Castagnetti et al. [93]	ima	559	6 y	NS	83% vs. 88% (*p* < 0.001)	MR4.0, 52% vs. 67% (*p* = 0.001)	At 7 y, 83% vs. 90% (*p* = 0.017)	PFS at 7 y, 81% vs. 89% (*p* = 0.005) FFS at 7 y, 54% vs. 71% (*p* < 0.001)
Pagnano et al. [95]	ima	170	6 y	At 6 mos, 43% vs. 70% (*p* = 0.02) At 12 mos, 62% vs. 78% (*p* = 0.16)	At 18 mos, 54% vs. 69% (*p* = 0.46)	NR	At 5 y, NS At 10 y, 93% vs. 73% (*p* = 0.03)	PFS and EFS, NS
Pfirrmann et al. [94]	ima	1494	6.5 y	NR	NR	NR	At 5 y, 89% vs. 93% (*p* = 0.02) ^a^	NR
Da Silva et al. [97]	ima	172	5 y	NR	NR	HR of e14a2 to e13a2 = 3.37 (95% CI 1.67–6.81; *p* = 0.001)	NR	NR
Sharma et al. [127]	ima	87	2 y	59% vs. 23% (*p* = 0.04) Considering only previously untreated pts, 61% vs. 35% (NS)	NR	NR	NR	NR
Vega-Ruiz et al. [103]	ima	480	5 y	NS	59% vs. 77% (*p* = 0.008) Post-IFN, 34% vs. 63% (*p* = 0.001)	25% vs. 47% (*p* = 0.002) Post-IFN, 16% vs. 42% (*p* = 0.001)	NR	EFS, NS TFS at 4 y, 93% vs. 98% (*p* = 0.08)
Polampalli et al. [128]	ima	202	1 y	NS	NS	NR	NR	NR
Mir et al. [99]	ima	200	NR	NR	64% vs. 72% (*p* = 0.04)	NR	NR	NR
Breccia et al. [98]	ima	208	7 y	NR	NR	sMR4.5, 31% vs. 43% (*p* = 0.02)	NR	NR
Rostami et al. [100]	ima	60	4 y	*p* = 0.02 in favour of e14a2	At 1 y, median of the BCR::ABL1 transcript ratio (%), 0.38 vs. 0.02	NR	NS	EFS, *p* = 0.03 in favour of e14a2
Nachi et al. [106]	ima	67	3 y	NR	At 18 mos, 50% vs. 77% (*p* = 0.09)	NS	NR	NR
Greenfield et al. [102]	ima	69	2.5 y	NR	At 1 y, 18% vs. 50%	NS	NS	EFS, NS
Marcé et al. [101]	ima	202	6 y	NS	At 6 mos, 9% vs. 18% (*p* = 0.088)	NS	95% vs. 83% (*p* = 0.022)	NR

The reported data include patients treated with frontline TKI expressing the e13a2 or e14a2 transcript (whether alone or in co-expression with the e13a2). *Pts* patients, *Ima* imatinib, *IFN* interferon, *FU* follow-up, *CCyR* complete cytogenetic response, *MMR* major molecular response (BCR-ABL1 ≤ 0.1%IS), *DMR* deep molecular response (MR4.0, BCR-ABL1 ≤ 0.01%IS; MR4.5, BCR-ABL1 ≤ 0.0032%IS), *sDMR* stable DMR, *sMR4.0* stable MR4.0, *sMR4.5* stable MR4.5, *Y* years, *Mos* months, *HR* Hazard Ratio, *CI* confidence interval, *OS* Overall Survival, *PFS* progression-free survival, *TFS* transformation-free survival, *EFS* event-free survival, *FFS* failure-free survival, *NS* not significant with *p* > 0.1, *NR* not reported. ^a^ In this study, when the log-rank test was stratified for the risk groups of the ELTS score, the significant survival advantage for patients with transcript e14a2 disappeared (*p* = 0.106).

**Table 4 cells-12-01703-t004:** Data on response to treatment and long-term outcomes according to transcript type in patients treated with different TKIs types, including newer-generation TKIs.

Ref.	TKI	No. of pts	FU	CCyR Rate e13a2 vs. e14a2	MMR Rate e13a2 vs. e14a2	DMR Rate e13a2 vs. e14a2	OS Rate e13a2 vs. e14a2	PFS/TFS/EFS/FFS Rate e13a2 vs. e14a2
Jain et al. [91]	ima, das, nilo	481	8 y	At 6 mos, 73% vs. 81%	At 1 y, 55% vs. 83%	At 12 mos, MR4.5 19% vs. 42% At 18 mos, MR4.5 28% vs. 50% At 60 mos, MR4.5 47% vs. 69%	NS	TFS at 5 y, 91% vs. 97% (*p* = 0.01) EFS at 5 y, 79% vs. 89% (*p* = 0.09); HR of e14a2 to e13a2 = 0.59 (95% CI 0.36–0.98; *p* = 0.04)
Sasaki et al. [108]	ima, das, nilo, pona	603	8.5 y	NS	At 1 y, 36% vs. 46%	sMR4.5, 34% vs. 45%	NR	NR
D’adda et al. [109]	ima, das, nilo	173	5 y	NS	NS	MR4.0, 52% vs. 82% (*p* = 0.008) sDMR, 27% vs. 47% (*p* = 0.004) ^a^	NR	NR
Shanmuganathan et al. [110]	NR	280	NR	NR	NR	At 6 y, MR4.5, 52% vs. 70%	NR	NR
Abdulla et al. [114]	ima, das, nilo, pona	79	2.5 y	NS	NS	NS	NR	NR
Su et al. [111]	ima, das, nilo	1124	4 y	At 1 y, cumulative incidence of CCyR: with ima, 45% vs. 59% (*p* = 0.001) with das, NS with nilo, 70% vs. 83% (*p* = 0.041)	At 1 y: ^b^ with ima, 27% vs. 38% (*p* = 0.010) with das, NS with nilo, 58% vs. 84% (*p* = 0.002)	At 1 y, NS (with any TKI)	At 5 y, NS (with any TKI)	PFS at 5 y, NS (with any TKI)
Genthon et al. [112]	nilo	118	4 y	NS	At 1 y, 50% vs. 67% (*p* = 0.048)	Cumulative incidence of MR4.5, 60% vs. 100% (*p* = 0.005)	NS	EFS, NS
Castagnetti et al. [113]	nilo	345	5 y	NR	NS ^c^	At 3 y, MR4.0, 56% vs. 66% (*p* = 0.06) ^c^	NS	PFS, NS
Jain et al. [116]	pona	85 (38 frontline, 47 RR)	NR	Among RR pts, 50% vs. 61%	Among RR pts, 29% vs. 52%	Among frontline pts, the median levels of transcripts at 3 and 6 mos, NS	At 3 y, among RR pts, 62% vs. 100% (*p* = 0.03)	Among RR pts, FFS at 3 y, 54% vs. 87% (*p* = 0.08)

The reported data include patients treated with frontline TKI expressing the e13a2 or e14a2 transcript (whether alone or in co-expression with the e13a2). *Pts* patients, *Ima* imatinib, *Das* dasatinib, *Nilo* nilotinib, *Pona* ponatinib, *FU* follow-up, *CCyR* complete cytogenetic response, *MMR* major molecular response (BCR-ABL1 ≤ 0.1%IS), *DMR* deep molecular response (MR4.0, BCR-ABL1 ≤ 0.01%IS; MR4.5, BCR-ABL1 ≤ 0.0032%IS), *sDMR* stable DMR, *sMR4.0* stable MR4.0, *sMR4.5* stable MR4.5, *Y* years, *Mos* months, *HR* Hazard Ratio, *CI* confidence interval, *OS* Overall Survival, *PFS* progression-free survival, *TFS* transformation-free survival, *EFS* event-free survival, *FFS* failure-free survival, *RR* relapsed/refractory, *NS* not significant with *p* > 0.1, *NR* not reported. ^a^ In this study, the achievement of a sDMR was significantly lower in patients treated with imatinib compared to those who received frontline 2GTKIs in both the e13a2 and e14a2 cohorts (*p* = 0.0485 and *p* = 0.0006, respectively). ^b^ In this study, the cumulative incidences of MMR at 12, 24, 36 and 60 months were significantly higher in patients treated with 2G-TKIs compared to those treated with imatinib in both the e13a2 and e14a2 cohorts (*p* < 0.01), in the absence of significant differences between nilotinib and dasatinib groups. ^c^ In this study, when grouping together the patients with the e14a2 transcript alone and those co-expressing both transcripts and comparing them to patients with e13a2 transcript alone, the response differences became significant (cumulative incidence of MMR and MR4.0, *p* = 0.050 and *p* = 0.036, respectively), in the absence of outcome differences (PFS and OS, *p* = 0.340 and *p* = 0.276, respectively).

### 2.5. Somatic Mutations

Although *BCR::ABL1* is the molecular hallmark driving CML pathogenesis, according to recent knowledge, CML is not a genetically uniform and single-hit disorder, but it is rather characterized by a complex genetic heterogeneity, still under investigation.

The emergence of point mutations within the kinase domain (KD) of *ABL1* is the most well-recognized mechanism of TKI resistance. However, approximately 40% of resistant cases are independent of BCR::ABL1 signaling and could be mediated by the accumulation of additional genomic aberrations, secondary to the genetic instability induced by the continuous unrestrained expression and activity of BCR::ABL1 kinase [129]. The accumulation of mutations in other genes in addition to *BCR::ABL1* has been associated with the disease progression to advanced phases. Moreover, some studies suggested that *BCR::ABL1* rearrangement is necessary, but it might not always be sufficient to induce CML, and other additional genetic events could contribute to the initiation of the disease itself [130,131]. Taking advantage of high-throughput sequencing techniques, extensive investigations have begun to uncover the mutational spectrum in other genes than *BCR::ABL1* in CML already at CP diagnosis.

About 35% of CML patients (ranging 25–50%) carry somatic mutations in cancer-associated genes at diagnosis [132,133,134]. The most frequently mutated gene at diagnosis in CP-CML patients (about 10%) is *ASXL1*; other genes recurrently mutated are *IKZF1* mutations and deletions, *RUNX1*, *DNMT3A*, *KMTD2*, *SETD1B*, *TET2*, *TP53* and *JAK2*. The majority of these are epigenetic regulator genes [132,133,135]. Several of these somatic mutations are regarded as preleukemic mutations and are mostly in CHIP (clonal hematopoiesis of intermediate potential)-related genes, including *DNMT3A*, *TET2* and *ASXL1* [136,137,138,139]. CHIP refers to an age-related abnormal clonal expansion of hematopoietic stem cells carrying somatically acquired mutations that confer a growth advantage (observed in around 10% of people older than 65 years lacking hematologic disorders) [140], which has been linked to an increased all-cause mortality, due mostly to a higher incidence of cardiovascular diseases, other than been associated with a slightly increased risk of hematologic malignancies [140,141]. However, the finding of *ASXL1* mutations in CML patients with a lower median age (<60 years) and at even higher frequency among children and young adults (29%), and, on the other hand, the identification of *DNMT3A* mutations at lower frequency in CML compared to CHIP and other myeloid neoplasms suggest that the acquisition of these mutations in CML is not an age- but a disease-related event [133,142,143]. At this stage, the role played by CHIP-related mutations in the initiation and mutational landscape of CML is still to be clarified [144].

Ongoing studies have focused on evaluating the potential clinical relevance of somatic mutations at CP-CML diagnosis. In general, a higher mutational burden in cancer-related genes at diagnosis is associated with poorer responses [137,145]. In particular, mutations in epigenetic modifiers seem to predict a worse short- and long-term response to TKI treatment, regardless of mutation clearance by the follow-up (FU) (see also the paragraph “Prognostic factors beyond the baseline”) [136,137,138,139]. In the study by Bidikian et al. [146], patients treated with imatinib or a 2G-TKI carrying *ASXL1* mutation at diagnosis showed significantly worse EFS and FFS compared to patients with no mutations. Schönfeld et al. [147] confirmed the worse prognostic impact of *ASXL1* mutations detected at diagnosis specifically in nilotinib-treated patients, associated with significantly lower MMR probabilities at 12, 18 and 24 months than patients without any mutations or mutations other than *ASXL1*.

## 3. Early Dynamic Risk Assessment: The Response to Treatment

According to current guidelines, the assessment of a “baseline” disease risk prior to initiation of TKI therapy, using clinical and hematologic data combined in the Sokal/ELTS prognostic systems, in addition to the cytogenetics data looking for the high-risk ACAs/Ph+, is recommended for all patients with CP-CML at diagnosis, playing a role in the choice of the appropriate treatment [18,19,20]. However, in the current era in which CML is typically a chronic disease with prolonged survival, determination of the patient risk status at diagnosis has a limited significance, and the response to TKIs is actually the most important prognostic factor. The monitoring of the response to treatment allows to assess the kinetics of *BCR::ABL1* transcript levels, determining whether the patient can continue the ongoing therapy or needs an intensification/switching of the treatment or may even be eligible for a TFR attempt [18,19,20,148,149].

The current recommended monitoring strategy in CML is based on molecular tests using standardized, validated and widely used RT-qPCR assays. Evidence supports the importance of the achievement of well-defined molecular response levels at specific time-points (3, 6, 12 and >12 months), regarded as molecular milestones, associated in cases of “optimal” response to the best long-term outcome, which is a CML-specific survival close to 100%. A “treatment failure/resistance” is a “red flag” entailing a treatment switch is warranted to limit the risk of progression and death [18,19,20,149]. Failure to achieve milestone responses defines the “primary resistance”, whereas a “secondary resistance” is defined by the loss of a previously achieved response [7].

In particular, the initial molecular response after 3 months of TKI therapy has been reported to have a prognostic significance: a failure to attain an early molecular response (EMR), defined by a *BCR::ABL1* transcript level < 10% on the International Scale (IS) [150] at 3 months, is associated with significantly inferior molecular response rates and OS and with an increased progression risk, irrespective of the TKI used [151,152,153,154]. Among cases who fail to meet standard EMR criteria, the rate of decline in *BCR::ABL1* transcript level from the individual patient baseline value when measured at 3 months is a significant and independent predictor of outcome, allowing the discrimination of the poorest-risk patients. Higher velocity of tumoral burden reduction has been associated with superior molecular response rates and long-term outcomes [155,156,157].

## 4. Prognostic Factors beyond the Baseline

In real-life experiences, about 25–45% of imatinib-treated patients switch to a 2G-TKI, and in 65–80% of cases the change is due to resistance [158,159,160]. Use of 2G-TKIs is associated with lower rates of resistance than imatinib, but still relevant: the reason for the switch after a frontline 2G-TKI, in clinical trial or real-life setting is treatment failure in about 10–40% of cases [15,161,162,163,164,165,166]. Prior failure is associated with a higher risk of a subsequent failure, with resistance rates increasing up to 50–65% in the second-line setting [160,167,168]. The molecular milestones to second-line treatment are the same as to first-line treatment [18], and, as for patients on frontline therapy, initial molecular response at 3 and 6 months on second-line therapy has predictive value [169,170,171,172]. CP-CML patients who are resistant to two or more TKIs or to just one 2G-TKI have limited therapeutic options and are at significant risk for progression to BP [173].

As for baseline, also in the context of second-line (and following) therapy, a number of variables have been tested as potential prognostic factors, and some combined within scoring systems proposed as prognostic models. In particular, the Hammersmith group [174] proposed a score aiming at predicting the sensitivity to a 2G-TKI (nilotinib or dasatinib) after imatinib-failure, based on three factors: the best cytogenetic response to imatinib, baseline Sokal score and recurrent neutropenia during imatinib treatment. The adverse prognostic implications of imatinib-induced myelosuppression had already been highlighted previously, hypothesized as a manifestation of a reduced residual normal stem cell pool and, therefore, a more aggressive disease [175,176,177]. The Hammersmith score was validated in further studies on larger scale and proved to have predictive value in terms of both cytogenetic/molecular responses and long-term outcomes [174,178,179]. On the other hand, so far, no prognostic models have been developed for patients after failure of 2G-TKIs in the first-line setting.

Cases of failure must be firstly investigated through a screening for *BCR::ABL1* kinase mutations [18,19,20]. Up to now, this mutational mechanism of resistance is the only “druggable” one, and the detection of the *BCR::ABL1* mutation status should guide the appropriate selection of the subsequent therapy to prevent the expansion of resistant clones under the inappropriate TKI [180]. Various studies have investigated the potential prognostic implications of *BCR::ABL1* KD mutations, that predominate in the events of secondary resistance (detectable in about 50–60% of cases) and become even more frequent in advanced phase disease (accounting for about 65–80% of AP/BP patients) [18,180,181]. Emergence of *ABL1* KD mutations during imatinib treatment is associated with a greater likelihood of resistance, progression and shorter survival [180,182]; patients with mutations in the adenosine triphosphate (ATP) binding loop (P-loop) have a particularly unfavorable prognosis [183,184]. The newer-generation TKIs have a much narrower spectra of mutations that confer a degree of insensitivity and have proved similar responses and outcomes for patients in CP with or without mutations [180,185]. Once the disease is in advanced phase, the mutation status has no significant impact on outcomes [185]. Both pre-existing mutations [186] and the sequential therapy with different TKIs [187] are associated with the emergence of additional mutations. Patients with multiple mutations have a worse prognosis than those with no or one mutation, especially those with “compound” mutations (as defined when they occur on the same *BCR::ABL1* molecule) conferring higher degrees of resistance involving multiple TKIs, including ponatinib [180,188,189,190].

Although sanger sequencing (SS) has long been considered the gold standard for mutation analysis, in the 2020 ELN recommendations, the recommended technology to detect *BCR::ABL1* resistance mutations in patients not responding adequately to TKIs is next-generation sequencing (NGS) [18] because of its greater sensitivity in identifying KD mutations, including low-levels mutations which seem to favor and predict clonal selection and disease progression [188,189,191,192].

Recent genomic studies have actually provided a more accurate picture of the heterogeneous and complex mutation landscape of the CML disease beyond *BCR::ABL1* throughout the disease course (see also the paragraph “Somatic mutations”). Frequency of detection and burden of somatic mutations other than *ABL1* KD mutations increase in cases of TKI-non response and, more prominently, with progression to advanced phases of CML, contributing to the disease transformation itself [132,133]. Over two-thirds of BP patients carry somatic mutations in one or more genes and, in addition to the most common *ABL1* KD mutations, the most frequently mutated genes in advanced phase patients are *RUNX1* (18–19%), *ASXL1* (15–17%) and *IKZF1* exon deletions (16–20%) [132,133]. Both studies by Grossman et al. [193] and Branford et al. [194] reported a very high frequency of co-occurrence of *ABL1* KD mutations and other mutated genes (85–90%), with the latter predated often the acquisition of the former. Multiple fusion genes with well-established leukemia-associated fusion partners have also been described in advanced phase CML (including *CBFB*, *RUNX1* and *KMTD2*), in addition to recurrent novel mutations (such as in *SETD2*, *SETD1B*, *BCOR* and *UBE2A* genes) that could have emerging roles as contributors to CML evolution [132,133].

Only a few studies have explored the dynamics of the mutational profile throughout the disease course in CML and its correlation with clinical outcomes. Kim et al. [136], performing a systematic analysis on matched diagnosis-FU samples from 100 CML patients (in the various phases of the disease, treated with imatinib or 2G-TKIs) and their sorted T-cell fractions as a representative of the hematopoietic stem cells fraction, identified five patterns of mutation dynamics under TKI therapy. Persistence at FU of mutations, mostly in genes involved in transcription, despite successful TKI response (pattern 1), suggested that these mutations of preleukemic nature existed independent of the Ph+ clone. The acquisition of new mutations (pattern 2), including *ABL1*, *TP53*, *KMTD2* and *TET2*, was strongly correlated with treatment failure, whereas patients exhibiting mutation clearance (pattern 3) showed mixed clinical outcomes. Patterns 4 and 5 were both characterized by the presence of preleukemic mutations, that persisted (pattern 4) or were cleared (pattern 5) at FU. Patterns 3 to 5 included mutations within epigenetic pathway genes (including *TET2* and *ASXL1*) that were associated with significantly lower responses to TKI therapy independent of mutation pattern 3, 4 or 5, that is regardless of mutation clearance during the disease course. Similar patterns of mutation dynamics were also described by two other longitudinal studies, both confirming in particular the association between the acquisition of new mutations and poor responses [132,133,137,194]. The persistence of preleukemic and Ph− mutations, involving mainly epigenetic modifier genes (*TET2* and *DNMT3A*), was associated with mixed responses [132,137]. Additional data are needed to gain a deeper understanding of the whole somatic genome in pathogenesis of CML and its impact on CML outcome.

In the multistep process of CML progression, cytogenetic clonal evolution is traditionally considered as a key phenomenon of disease transformation and an indicator of poor prognosis. Overall, patients with emergence of two or more ACAs simultaneously have a worse survival than patients with single ACAs. When compared with patients with no ACAs, patients with ACAs (even a single ACA) generally show a worse survival [61]. However, not all ACAs acquired during therapy predict survival equally [18,195]. Wang et al. [196] showed that isolated +8 had a favorable impact with better survival in comparison to cases with other single ACAs, whereas some minor route changes such as 3q26.2 abnormalities or −7/7q- conferred dismal responses to TKI treatment and high risk of disease progression with poorer OS compared with other ACAs [197]. On the other hand, there was no significant difference in survival between patients with −Y and patients with no ACAs [61]. Recently, Gong et al. [75] stratified 2326 patients into four risk groups, based on risk of blastic transformation associated with each ACA: the standard-risk (SR) group included patients without ACAs; the high-risk (HR) group included patients with i(17q), −7/del(7q) and 3q26.2 rearrangements isolated or as components of complex ACAs; the intermediate-1 (Int-1) risk group included patients with +8, (+Ph) or other single ACAs; and the intermediate-2 (Int-2) risk group included patients with other complex ACAs. This risk stratification correlated well with the patient survival: the 8-year OS was 79.7%, 57.6%, 47.0% and 31.2% for the SR, Int-1, Int-2 and HR groups, respectively. The impact of ACAs on blastic transformation proved to be partially lineage-specific: all ACAs showed a variable degree of risk of progression to myeloid BP, whereas only −7/7q- carried a significant risk of lymphoid blastic transformation. The concurrent presence of ACAs and hematologic AP (in terms of increased blast count) correlates with a further increased risk of blastic transformation, significantly with higher-risk ACAs [75,198], but the impact of high-risk ACAs is greater at the lowest blast counts: according to the German CML Study Group [77], at blast levels of 1–5%, high-risk ACAs showed an increased hazard to die compared to no ACAs in contrast to low-risk ACAs; the impact of high-risk ACAs decreased with blast increase up to 15%, whereas no significative differences between patients with and without high-risk ACAs was observed at blast levels of at least 20%. This also means that the prognostic value of ACAs depends not only on the type of ACA but also on the phase of occurrence: once the disease has progressed to BP, the subsequent outcome becomes substantially independent of the kind or the complexity of ACAs [199].

In all studies investigating the role of cytogenetic abnormalities in CML, only those detected in Ph+ cells have been considered ACAs. As to the emergence of clonal chromosomal abnormalities in the Ph− metaphases (CCA/Ph−) during TKI treatment, the little available data have not provided convincing results regarding their prognostic significance [200]. In a recent retrospective study by Issa et al. [200,201], among the 598 evaluated patients with newly diagnosed CP-CML treated with various TKIs, CCA/Ph− occurred in 58 patients (10%), with −Y as the most common. They found no significant difference in molecular response rates between patients with CCA/Ph− and the control group of patients without any ACA (neither in the Ph+ nor in the Ph− metaphases). However, the former had a significantly decreased survival than the latter. In particular, patients with non −Y CCA/Ph− had the worst comparative long-term outcomes compared with both patients without ACAs and patients with −Y as the sole CCA/Ph−. The presence of −Y as the only CCA/Ph− was associated with survival rates lower than the control group without ACAs, though not statistically significant.

## 5. Summary and Conclusions

In the present scenario of CML, marked by the remarkable success of the lifelong oral therapy with TKIs and in which some patients may even aim for an “operational cure” (that is prolonged survival in molecular remission even in presence of residual quiescent leukemic stem cells) off-treatment (TFR) [18,202,203], occurrence of advanced phase still remains a major challenge with limited effective therapeutic options and a prognosis not much better than that after conventional therapy. On the other hand, the low progression rates under TKIs indicate that blast crisis can be prevented [21]. It is therefore essential to identify predictive tools for a prompt recognition of those patients at higher risk of progression and CML-induced death, worthy of closer molecular monitoring and treatment intensifications—considering potentially the choice of a frontline 2G-TKI or the early switch to a more powerful TKI in following lines or even evaluating the eligibility to an allo-SCT or new agents [204]. This is also the rationale for the new 2022 World Health Organization (WHO) classification for myeloid neoplasms in which the “CML-AP” as diagnostic category is omitted, putting emphasis on the identification of the high-risk features that, regardless of the historical criteria of AP, are associated with CP progression and poor outcome [9].

Initially, the research focused on clinical and hematologic data collected at diagnosis of the disease, introducing the prognostic scoring systems developed in the various therapeutic eras. Although based on simple parameters, they are still a fundamental tool for predicting disease risk and planning the therapeutic strategy, especially the most-widely used Sokal score and the internationally recommended ELTS score.

In the 1970s, when the structure of the Ph chromosome was defined, the analysis of the chromosome banding pattern of BP-CML patients demonstrated the presence of additional non-random chromosome aberrations [66]. Further studies provided insights into the role of ACAs on disease progression in CML and their impact on survival. Current evidence [195] indicates that this relationship between ACAs and CML outcome is far from being uniform: the prognostic power of ACAs differs based on the specific type of abnormality, the disease stage, the modality of occurrence (alone or in combination) and the time of occurrence (early at diagnosis or later on). CML-CP patients with ACAs recognized as at high-risk, whether emerging at or after diagnosis, have a worse survival [195]. Data indicate also low-risk ACAs may have a negative impact on survival compared to no ACA, although these results often proved not significant [195]. The emergence of high-risk ACAs in the presence of low blast levels appears to be an earlier indicator of CML-related death than standard blast thresholds, suggesting the requirement of an intensification of the treatment before a further increase of blasts [77].

Another field of investigation is represented by the potential prognostic impact of the various *BCR::ABL1* transcript types [205]. The e13a2 transcript negatively affects the rate, depth and speed of responses to imatinib, and patients with this type of transcript may benefit from a 2G-TKI as frontline therapy. However, according to most reported data, the type of transcript does not seem to affect long-term outcomes, regardless of the TKI. The higher probability in achieving and maintaining a DMR associated with e14a2 indicates that the identification of the type of transcript at baseline might have an impact on eligibility for TFR, but data are inconsistent. The influence of the *BCR::ABL1* transcript type in CML still remains controversial, also due to technical bias, which could be bypassed by turning into alternative technologies, such as digital PCR (dPCR). Further studies are warranted to clarify this issue [124]. Currently, prognostic scores and high-risk ACAs are the only widely used and validated factors for risk stratification of CP-CML patients at baseline [18].

More recently, there has been a growing focus on the molecular/biological aspects of CML disease. Much larger studies are required to assess the risk conferred by mutated cancer genes at diagnosis and on treatment. The integration of genomic data in risk stratification represents a promising strategy to enable more precise identification of high-risk patients. The need to elucidate the genomic events that underlie the disease course has laid the foundation for the recent HARMONY CML research project [206] with the interest to deepen the knowledge of biomarkers as players in CML pathogenesis, predictive factors of treatment response and outcome and potential targets for novel therapeutic approaches. The project is based on “Big Data” of the HARMONY Platform, a shared database of mutational data pool collected from several countries (European and non). One of the interesting molecular aspects of CML concerns the potential association between CHIP mutations and the development or exacerbation of cardiovascular events during TKI treatment. A recent study by Hadzijusufovic et al. [207] reported a significantly higher frequency of CHIP mutations in nilotinib-treated patients who developed arterial occlusive disease (AOD) compared to those without AOD. Given the association of the newer-generation TKIs with cardiovascular diseases (especially in patients with pre-existing risk factors) [37,38,208] and the prognostic impact of comorbidities on CML survival, a better definition of the role of CHIP mutations in CML is worthy of further exploration for an improved management of the lifelong TKI algorithm.

A natural consequence of the overall success of CML therapy, with a growing number of patients having a near-normal life expectancy, has been a necessary increasing attention to the long-term outcomes other than the quality of life of these CML “survivors”. Consequently, a tailored approach to the CML patient requires the consideration of the adverse events associated with the TKIs as well, including the side-effects, since non-adherence to therapy is a key treatment-failure risk factor and a real challenging clinical issue [209,210,211,212].

In conclusion, predicting response in CML patients is vitally important to succeed in the primary goal of maintaining patients in chronic CML. Recent technological advances and the collection of increasingly data are enabling the gaining of significant insights into the mechanisms that underlie disease transformation. In this way, we are identifying new potential biomarkers, originating in the highly proliferative and genetically unstable CML cells, through the investigation and a better definition of the role of each ACA and somatic mutations, also according to the emerging phase and time in the CML disease. The future direction of this research involves the incorporation of these newly emerging factors to those already validated in disease risk assessment, response monitoring and therapeutic decisions for an optimized management of individual CML patients.

## Figures and Tables

**Table 1 cells-12-01703-t001:** Diagnostic criteria for accelerated phase (AP) and blast phase (BP) chronic myeloid leukemia (CML) according to the major classification systems used in clinical trials and practice.

	ELN (2020)	WHO (2016)	WHO (2022)
**Accelerated phase**	PB or BM blasts 15–29% or blasts plus promyelocytes > 30% with blasts <30%	PB or BM blasts 10–19%	// ^a^
	PB basophils ≥20%	PB basophils ≥ 20%	
	Platelets <100 × 10^9^/L unrelated to therapy	Platelets <100 × 10^9^/L unrelated to therapy or >1000 × 10^9^/L unresponsive to therapy	
		Increasing spleen size and/or WBC count unresponsive to therapy	
	CCA/Ph+ on treatment	CCA/Ph+ on treatment; CCA/Ph+ major route, complex karyotype or 3q26.2 abnormalities at diagnosis	
**Blast phase**	PB or BM blasts ≥30%	PB or BM blasts ≥20%	PB or BM myeloid blasts ≥20%
	Extramedullary blast proliferation	Extramedullary blast proliferation	Extramedullary blast proliferation
			Presence of increased PB or BM lymphoblasts

*ELN* European LeukemiaNet, *WHO* World Health Organization, *PB* peripheral blood, *BM* bone marrow, *WBC* white blood cells, *CCA* clonal chromosomal abnormalities, *Ph+* Philadelphia chromosome positive. ^a^ Notably, the most recent 2022 WHO classification for myeloid neoplasm has omitted “AP” as a diagnostic category and CML phases have been consolidated into chronic and blast phases only. In this review, we use the terms “AP” and “BP” as historically defined by the previous classification systems.

**Table 2 cells-12-01703-t002:** Baseline prognostic models in chronic myeloid leukemia (CML).

Score System	Patient Population ^a^	Endpoint	Calculation	Risk Groups (Proportion)
Sokal score [46]	678 pts treated with chemotherapy (in a minority of cases associated with splenectomy or immunotherapy), diagnosed in 1962–1978	Probabilities of survival	Exp (0.0116 × (age [years] − 43.4)) + 0.0345 × (spleen size [cm] − 7.51) + 0.1880 × ((platelet count [10^9^/L]/700)^2^ − 0.563) + 0.0887 × (peripheral blood blasts [%] − 2.10)	Low-risk: <0.80 (39%) Intermediate-risk: ≥0.80 and ≤1.20 (38%) High-risk: >1.20 (23%)
Euro (Hasford) score [47]	908 pts treated with IFN-α (either alone or in combination with chemotherapy), diagnosed in 1983–1994	Probabilities of survival	(0.6666 × age [0 when <50 years, 1 otherwise] + 0.0420 × spleen size [cm] + 0.0584 × peripheral blood blast [%] + 0.0413 × eosinophils [%] + 0.2039 × basophils [%] [0 when <3%, 1 otherwise] + 1.0956 × platelet count [0 when <1500 × 10^9^/L, 1 otherwise]) × 1000	Low-risk: ≤780 (40.6%) Intermediate-risk: >780 and ≤1480 (44.7%) High-risk: >1480 (14.6%)
EUTOS score [48]	926 pts treated with imatinib, diagnosed in 2002–2006	Probabilities of CCyR at 18 months	7 × basophils [%] + 4 × spleen size [cm]	Low-risk: ≤87 (90%) High-risk: >87 (10%)
ELTS score [49]	2205 pts treated with imatinib, diagnosed in 2002–2006	Probabilities of dying of CML	0.0025 × (age [years]/10)^3^ + 0.0615 × spleen size [cm] + 0.1052 × peripheral blood blasts [%] + 0.4104 × (platelet count [10^9^/L]/1000)^−0.5^	Low-risk: ≤1.5680 (61%) Intermediate-risk: >1.5680 and ≤2.2185 (27%) High-risk: >2.2185 (12%)

*Pts* patients, *IFN* interferon, *EUTOS* European Treatment and Outcome study, *CCyR* complete cytogenetic response, *ELTS* EUTOS Long Term Survival. ^a^ Number of patients with complete data used for the final estimation of the regression coefficients.

## Data Availability

Not applicable.

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
