# Peer review of "Prognosis in Chronic Myeloid Leukemia: Baseline Factors, Dynamic Risk Assessment and Novel Insights"

_cells, 2023, doi:10.3390/cells12131703_

Round 1

Reviewer 1 Report

The review entitled: “Prognosis in Chronic Myeloid Leukemia: baseline factors, dynamic risk assessment and novel insight” (cells-2444246) by Iezza et.al. summarizes current prognostic factors, most appropriate TKI on the basis of the profile and aggressiveness in patients with CP-CML.

Albeit the review is well written, prepared and of special interest, comments should be addressed to further improve the manuscript.

Comments:

1.    The whole manuscript until point 3 should shortened to avoid repetition given in the text versus tables.

2.    Section 4, page 14, line 498-546: this section should be also shortened.

3.    Section 5, page 15: The term: “operational cure” should be adapted/changed or more defined within this part.

4.    Section 5, page 16: the authors should also add more information about what are exact new insights of the review. What can be learnt for the clinical practice and what could be the recommendation according most appropriate TKI within different disease profiles given from this review.

5.    Page 16: Please add author’s contribution and funding.

6.    Please check the manuscript according abbreviations and whether they were defined from the first time of introduction.

Reviewer 2 Report

This is an excellent review with special emphasis on cytogenetic and genetic factors influencing prognosis in CML. The following points need consideration:

- Line 172: ACAs were named major and minor route according to their frequency in blastic phase. Please add the phase in which the frequencies of ACAs defined major/minor route.

- Line 348 replace HZ by HR.

- Somatic ASXL1 mutations could impact prognosis of CML in children and AYAs. Since they occur more frequently in children than in adults, please add a mention on ASXL1 mutations in the paragraph 90-96.

- ELTS is the only score that can be applied in children. Please make a mention of that.

- Last but not least, funding and COI should be mentioned in the version submitted for revision.

Round 2

Reviewer 1 Report

The review entitled: “Prognosis in Chronic Myeloid Leukemia: baseline factors, dynamic risk assessment and novel insight” (cells-2444246) by Iezza et.al. summarizes current prognostic factors, most appropriate TKI on the basis of the profile and aggressiveness in patients with CP-CML.

The authors addressed all my initial comments adequately.